# Learn from the Past: Dynamic Data Pruning with Historically Weighted Bernoulli Sampling

## Abstract

Dynamic data pruning, also known as data importance sampling, has been proposed to improve the efficiency of training large machine learning models. In the case of sampling with replacement, the optimal sampling distribution to minimize the variance is to sample proportionally to the gradient norm. However, this could result in repeated samples, which is an undesirable property. Noticing that most dynamic data pruning methods that avoid repeated samples can be viewed as weighted Bernoulli sampling, in this work, we derive the optimal distribution to minimize variance, thereby accelerating convergence. Furthermore, to eliminate the need for an extra computation for gradient estimation, we study the use of historical statistics. We then propose Omission (**O**ptimal inspired historical weighted Bernoulli sampling), which tracks the exponential moving average of the squared gradient norm and adopts scale normalization as well as probability smoothing to improve the performance. Experiments on image classification and large language model instruction finetuning tasks show that our proposed method achieves higher accuracy than other dynamic pruning methods.

## 1 Introduction

The substantial growth in dataset sizes has made training large machine learning models a challenging and time-consuming task. However, it is crucial to recognize that not all samples contribute equally to the learning process, and more importantly, the relevance of samples can change dynamically during training. Some samples might be initially important but become less informative as the model learns. This observation has motivated the field of dynamic data pruning, also known as data importance sampling, which aims to enhance the training efficiency of machine learning models (Alain et al., 2015; Zhao & Zhang, 2015; Katharopoulos & Fleuret, 2018). This technique dynamically assigns non-uniform sampling probabilities to each sample, with the goal of reducing variance in the gradient estimator and thereby accelerating the minimization of the loss function. In the case of sampling with replacement, it has been shown that the optimal sampling weight distribution for minimizing variance is proportional to the gradient norm of each sample (Alain et al., 2015; Zhao & Zhang, 2015). However, directly computing the full per-sample gradient to obtain the norm defeats the purpose of importance sampling. Therefore, several works approximate gradient norm using alternative approaches, such as focusing on the gradient of the final layer (Katharopoulos & Fleuret, 2018).

Despite the theoretical justification, a notable drawback of sampling with replacement is its potential to generate repeated samples, which can be an undesirable property. Instead, researchers have started to propose other dynamic data pruning methods that avoid repeated samples (Jiang et al., 2019; Raju et al., 2021; Qin et al., 2023; Thao Nguyen et al., 2024). We observe that most of these methods can be broadly categorized as weighted Bernoulli sampling. However, the sampling weights in these methods are typically determined by some heuristics, not through variance minimization. This limitation motivates our research to find optimal sampling weights for weighted Bernoulli sampling. We prove that the sampling probability should be proportional to the gradient norm for samples with smaller gradients, while being capped at one for those with larger gradients. Furthermore, for the case where there are random variations caused by drop-out or data augmentation, we conclude that the sampling probability should be linear to the square root of the expected gradient norm squared or capped at one.

Furthermore, to avoid additional forward passes on unselected samples, we study the use of historical statistics to approximate the gradient norm at the current step. We then propose Omission (**O**ptimal **ins**pired hi**st**orical we**i**ghted Bern**o**ulli sampli**ng**). Inspired by the optimal sampling probability distribution under random variation, we maintain an exponential moving average of the squared gradient norm for each sample. To enhance performance, we also normalize the scale by tracking the exponential moving average across all the samples and conduct probability smoothing to improve the performance.

We conduct extensive experiments on several image classification and large language model instruction finetuning tasks. The results show that our proposed method achieves higher accuracy than other dynamic pruning methods.

Our contribution can be summarized as follows.

- We derive the optimal distribution to minimize variance in weighted Bernoulli sampling, thereby accelerating convergence.
- Inspired by the optimal distribution, we propose Omission (**O**ptimal **ins**pired historical we**i**ghted Bern**o**ulli sampli**ng**), which utilizes historical statistics to avoid additional forward passes on unselected samples
- We propose to track the exponential moving average of the squared gradient norm and adopt scale normalization as well as probability smoothing to improve the performance.
- Our results show consistent improvements against state-of-the-art dynamic data pruning algorithms.

## 2 BACKGROUND

We consider training a deep neural network that takes an input vector $\boldsymbol{x}$ and produces an output $f(\boldsymbol{\theta}, \boldsymbol{x})$. With a training set $\{(\boldsymbol{y}_i, \boldsymbol{x}_i)\}_{i=1}^n$ consisting of $n$ samples and a loss function $\ell$, the network parameters $\boldsymbol{\theta}$ are learned by minimizing the empirical loss $\mathcal{L}$ over the training set:

$$\min_{\boldsymbol{\theta}} \mathcal{L}(\boldsymbol{\theta}) := \frac{1}{n} \sum_{i=1}^n \ell_i(\boldsymbol{\theta}),$$

where

$$\ell_i(\boldsymbol{\theta}) \equiv \ell(f(\boldsymbol{\theta}, \boldsymbol{x}_i), \boldsymbol{y}_i).$$

Stochastic optimizers (such as SGD and Adam (Kingma & Ba, 2014)) are widely used for solving this optimization problem for neural network training, where one or a batch of samples is selected at each step for gradient estimation. This paper studies how to conduct data importance sampling in stochastic optimizers.

At each step $t$, a total of $b$ samples are sampled to generate a gradient estimate $\hat{\mathbf{g}}_t$ to approximate $\nabla\mathcal{L}(\boldsymbol{\theta}_t)$. For any gradient estimation that is unbiased

$$\mathbb{E}[\hat{\mathbf{g}}_t] = \nabla\mathcal{L}(\boldsymbol{\theta}_t),$$

its variance can be associated with the reduction of the loss function as follows.

Assume the loss function is $L$-smooth, we have

$$\mathcal{L}(\boldsymbol{\theta}_{t+1}) = \mathcal{L}(\boldsymbol{\theta}_t - \alpha_t\hat{\mathbf{g}}_t) \leq \mathcal{L}(\boldsymbol{\theta}_t) - \alpha_t\nabla\mathcal{L}(\boldsymbol{\theta}_t)^\top\hat{\mathbf{g}}_t + \frac{\alpha_t^2 L}{2}\|\hat{\mathbf{g}}_t\|_2^2. \tag{1}$$

Define the variance of a vector as the sum of the variances of its individual coordinates

$$\mathrm{Var}(\boldsymbol{x}) \equiv \sum_j \mathrm{Var}(x_j),$$

we can then establish a connection between the reduction in the loss function and the variance of gradient estimation by taking the expectation of both sides of equation 1:

$$\mathbb{E}[\mathcal{L}(\boldsymbol{\theta}_{t+1})] \leq \mathcal{L}(\boldsymbol{\theta}_t) - \alpha_t\nabla\mathcal{L}(\boldsymbol{\theta}_t)^\top\mathbb{E}[\hat{\mathbf{g}}_t] + \frac{\alpha_t^2 L}{2}\mathbb{E}[\|\hat{\mathbf{g}}_t\|_2^2]$$

$$= \mathcal{L}(\boldsymbol{\theta}_t) - \alpha_t\|\nabla\mathcal{L}(\boldsymbol{\theta}_t)\|_2^2 + \frac{\alpha_t^2 L}{2}(\mathrm{Var}(\hat{\mathbf{g}}_t) + \|\mathbb{E}[\hat{\mathbf{g}}_t]\|_2^2) \tag{2}$$

$$= \mathcal{L}(\boldsymbol{\theta}_t) - (\alpha_t - \frac{\alpha_t^2 L}{2})\|\nabla\mathcal{L}(\boldsymbol{\theta}_t)\|_2^2 + \frac{\alpha_t^2 L}{2}\mathrm{Var}(\hat{\mathbf{g}}_t).$$

Consequently, we should design the sampling distribution to minimize the variance. In the case of sampling with replacement, this leads to an optimal sampling distribution where the sampling probability of each sample is proportional to its gradient norm $\|\nabla \ell_i(\boldsymbol{\theta}_t)\|$ (Alain et al., 2015; Zhao & Zhang, 2015). Since directly calculating $\nabla \ell_i(\boldsymbol{\theta}_t)$ defeats the purpose of importance sampling, Katharopoulos & Fleuret (2018) propose to use the gradient norm of the logits as an approximation, which still requires a forward pass on unselected samples.

Despite the theoretical justification, sampling with replacement could result in repeated samples, which can be undesirable. Instead, researchers have started to propose other dynamic data pruning methods that avoid repeated samples. Jiang et al. (2019) propose to use the forward pass to obtain the loss of each sample and conduct Bernoulli sampling, where the sampling probability of each sample is determined by the ranking of its loss. Qin et al. (2023) propose to use the previous loss function value as the score to eliminate the additional forward pass. Samples with scores larger than the average are selected with probability 1, while the rest are selected with a discounted probability $r$. Raju et al. (2021) propose to maintain a moving average of the previous loss function value as the score. A fixed fraction of $F$ lowest score samples are selected with probability $\epsilon$, while the rest are selected with probability 1. Thao Nguyen et al. (2024) propose to prune a fraction of $F_i$ of the lowest-loss samples, where $F_i$ is determined by the epoch number $i$.

These methods can all be roughly viewed as weighted Bernoulli sampling, as samples do not repeat and are selected independently. However, the sampling weights in these methods are all determined by some heuristics, not through variance minimization. This limitation motivates our research to find optimal sampling weights for weighted Bernoulli sampling.

## 3 PROPOSED ALGORITHM

In this work, we consider weighted Bernoulli sampling, which assigns a sampling weight to each sample and selects them independently, thus eliminating repeated samples. We then derive the optimal sampling strategy for this method. Even though Bernoulli sampling can lead to variation in the number of samples selected per step, in practice we observe that this can be approximated by keep sampling until a fixed batch size is reached, which will be discussed more in Section 3.3.

In this section, we start by introducing the unbiased formulation for weighted Bernoulli sampling. Given a larger batch of size $B$, we employ weighted Bernoulli sampling to select a smaller batch of size $b$ to approximate $\nabla \mathcal{L}(\boldsymbol{\theta}_t)$. For brevity, we omit the subscript $t$ from this point forward. Let $p_i$ be the probability of sampling the $i$-th sample in the larger batch, $\boldsymbol{g}_i$ be the sample gradient, and $\mathrm{r}_i$ be the random variable to indicate whether the sample is selected. We have

$$\sum_{i=1}^{B} p_i = b. \tag{3}$$

If we have $p_i > 0$ for all $i$, the following gives us an unbiased estimation

$$\frac{1}{B} \sum_{i=1}^{B} \mathbb{1}\{\mathrm{r}_i = 1\} \frac{\boldsymbol{g}_i}{p_i}. \tag{4}$$

Note that although several previous works also consider Bernoulli sampling, their estimation is often biased. For example, Thao Nguyen et al. (2024) have $p_i = 0$ for some samples and thus the method is biased; Jiang et al. (2019) and Raju et al. (2021) choose not to divide $p_i$ and thus give a biased estimation; Qin et al. (2023) divide by $b$ instead of $B$, which effectively increases the learning rate by $B/b$.

### 3.1 OPTIMAL SAMPLING WEIGHT TO MINIMIZE VARIANCE

As mentioned in Section 2, the variance of the gradient estimation $\mathrm{Var}(\hat{\mathbf{g}}_t)$ is connected with the reduction of the loss function. We now derive the optimal sampling weights that minimize variance for weighted Bernoulli sampling.

The derivation is as follows. Given the independence of each Bernoulli sample, the variance is simply the sum of the variances of the individual samples. For the $i$-th sample, the variance is given by

$$p_i(1 - p_i)\frac{\|\boldsymbol{g}_i\|^2}{B^2 p_i^2} = (1 - p_i)\frac{\|\boldsymbol{g}_i\|^2}{B^2 p_i}.$$

Taking the derivative of this expression, we obtain $-\|\boldsymbol{g}_i\|^2/(B^2 p_i^2)$, whose absolute value is monotonically decreasing. This indicates that the marginal benefit of reducing variance is decreasing as $p_i$ increases.

To achieve minimal variance, the marginal value of reducing variance should be equal across all samples, except for those with a probability of 1. Consequently, for a fixed budget $\sum p_i = b$, the optimal probability allocation is that some samples with the larger $\|\boldsymbol{g}_i\|$ have the selection probability 1, while the others have selection probability proportion to $\|\boldsymbol{g}_i\|$, similar to the case of sampling with replacement.

Therefore, the optimal distribution can be written formally as the following theorem.

**Theorem 1** *Given $\|\boldsymbol{g}_i\|$, to obtain the minimal variance for a fixed budget $b$, the sampling probability $p_i$ should satisfy*

$$p_i = \min(\frac{\|\boldsymbol{g}_i\|}{c}, 1),$$

*where $c$ is a constant across all samples and is determined by the constraint $\sum_{i=1}^{B} p_i = b$.*

In practice, we can use binary search to efficiently find the value of $c$. This result can also be easily extended to the case where $\boldsymbol{g}_i$ is non-deterministic under some random variation like drop-out or data augmentation.

**Theorem 2** *Given the expectation $\mathbb{E}[\|\boldsymbol{g}_i\|^2]$ across the model's random variation and data augmentation, to obtain the minimal variance for a fixed budget $b$, the sampling probability $p_i$ should satisfy*

$$p_i = \min(\frac{\sqrt{\mathbb{E}[\|\boldsymbol{g}_i\|^2]}}{c}, 1),$$

*where $c$ is a constant across all samples and is determined by the constraint $\sum_{i=1}^{B} p_i = b$.*

We provide detailed proof in the appendix.

## 3.2 Utilizing Historical Statistics

In practice, computing $\|\boldsymbol{g}_i\|$ at the current step defeats the purpose of data importance sampling. One common way to address this is to utilize the gradient of the logits

$$\tilde{\boldsymbol{g}}_i \equiv \|\frac{\partial\ell(\boldsymbol{z}, \boldsymbol{y}_i)}{\partial\boldsymbol{z}}\|\Big|_{\boldsymbol{z}=f(\boldsymbol{\theta}, \boldsymbol{x}_i)},$$

which can be computed in time linear to the number of logits (Katharopoulos & Fleuret, 2018; Jiang et al., 2019). However, this still requires extra forward passes for the unselected samples.

To avoid the extra forward passes, some work uses historical statistics. Specifically, Infobatch (Qin et al., 2023) and KAKURENBO (Thao Nguyen et al., 2024) utilize the last loss value for the sample. However, the randomness inherent in drop-out and data augmentation can cause this value to greatly deviate from the true value.

### 3.2.1 Exponential Moving Average of Gradient Norm Squared

To address the random noise in the historical statistics, we propose to use the exponential moving average with scale normalization for smoothing.

Inspired by Theorem 2, we propose to track the exponential moving average of $\|\tilde{\boldsymbol{g}}_i\|^2$. However, the scale of the gradient can be much larger for the earlier epochs, making them dominating the average.

To account for the change in gradient scale across epochs, we propose tracking the exponential moving average of the expectation $\mathbb{E}_i[\|\tilde{\boldsymbol{g}}_i\|^2]$ to conduct normalization.

Specifically, we propose the following formulation

$$\overline{\text{SCORE}}_i = \frac{\|\tilde{\boldsymbol{g}}_i\|^2}{\text{EMA}[\mathbb{E}_j[\|\tilde{\boldsymbol{g}}_j\|^2]; \tilde{\beta}_2]}$$

The expectation over $j$ in the denominator computes the exponential moving average of gradient norm squared across all the samples. This serves the purpose of normalizing the scale of the gradient.

We then take another exponential moving average

$$\text{SCORE}_i = \text{EMA}[\overline{\text{SCORE}}_i; \tilde{\beta}_1]$$

to compute the score for each sample, where $\text{EMA}[x; \beta]$ calculates the exponential moving average of $x$ with weight factor $\beta$. To summarize, the score of a sample is its gradient norm squared normalized by the average of all the samples. We then conduct an exponential moving average on the score. Without any tuning, we select $\tilde{\beta}_1 = 0.9$, $\tilde{\beta}_2 = 0.99$. following the commonly used parameters in Adam.

It is worth noticing that since we are using historical statistics, it is actually possible to use the full per-sample gradient $\boldsymbol{g}_i$ as we are already doing backward on these samples to obtain the full gradient. However, in our experiments, we observed that calculating the full per-sample gradient incurs a non-negligible overhead compared to calculating the full gradient, with no observable gain in the model's accuracy. Therefore, we still choose to use the gradient of the logits $\tilde{\boldsymbol{g}}_i$.

### 3.2.2 Score Smoothing

Revisiting the unbiased formulation of weight Bernoulli sampling

$$\frac{1}{B}\sum_{i=1}^{B}\mathbb{1}\{\mathrm{r}_i = 1\}\frac{\boldsymbol{g}_i}{p_i},$$

we can see that the term $\boldsymbol{g}_i/p_i$ could potentially be very large when $p_i$ is small. This is especially true when our score estimation is inaccurate, assigning a small $p_i$ to a large $\boldsymbol{g}_i$. This could be the potential reason that Jiang et al. (2019) and Raju et al. (2021) choose not to divide $p_i$, which leads to a biased estimation.

Instead, we propose score smoothing to address this issue. Specifically, we propose the following as the sampling probability.

$$p_i = \min\left(\frac{\sqrt{(1-\alpha)\text{SCORE}_i + \alpha\mathbb{E}_j[\text{SCORE}_j]}}{c}, 1\right), \tag{5}$$

where $c$ is determined by the constraint $\sum_{i=1}^{B} p_i = b$ and $0 \leq \alpha \leq 1$ is the smoothing factor. In the experiments, we choose $\alpha = 0.1$, similar to what is commonly used for label smoothing.. We use binary search to efficiently find the $c$ to satisfy the budget constraint.

From the budget constraint, we have

$$\sum_{i=1}^{B}\frac{\sqrt{(1-\alpha)\text{SCORE}_i + \alpha\mathbb{E}_j[\text{SCORE}_j]}}{c} \geq b. \tag{6}$$

Since $\mathbb{E}[x] \leq \sqrt{\mathbb{E}[x^2]}$, this gives us

$$\mathbb{E}_i[\sqrt{(1-\alpha)\text{SCORE}_i + \alpha\mathbb{E}_j[\text{SCORE}_j]}] \leq \sqrt{\mathbb{E}_i[(1-\alpha)\text{SCORE}_i + \alpha\mathbb{E}_j[\text{SCORE}_j]]}$$

$$=\sqrt{(1-\alpha)\mathbb{E}_i[\text{SCORE}_i] + \alpha\mathbb{E}_j[\text{SCORE}_j]} = \sqrt{\mathbb{E}_j[\text{SCORE}_j]}$$

Consequently, by equation 6,

$$c \leq \frac{B}{b}\sum_{i=1}^{B}\frac{\sqrt{(1-\alpha)\text{SCORE}_i + \alpha\mathbb{E}_j[\text{SCORE}_j]}}{B} \leq \frac{B}{b}\sqrt{\mathbb{E}_j[\text{SCORE}_j]}.$$

Combined with equation 5, we have

$$p_i \geq \frac{b}{B}\sqrt{\alpha},$$

which ensures the sampling probability is not too small for all the samples.

### 3.3 IMPLEMENTATION DETAILS

For our proposed methods, we provide two different implementations. The first implementation strictly follows equation 4, which obtains a larger batch $B$ through random shuffling. We then use weighted sampling to select a batch of size $b$. This has a variable number of samples at each step with an expected value of $b$. The second implementation is an approximation to the first one based on the observation that, unlike the method of Katharopoulos & Fleuret (2018) and Jiang et al. (2019), which relies on the statistics of the current step, our score relies on historical statistics, which is determined at the beginning of the epoch. Thus, we can instead sample from all of the $n$ training samples and then randomly group them into batches of size $b$.

We observe little performance differences for the two implementations, so the second implementation might be preferable in practice, as it produces a fixed batch size while being easier to implement.

### 3.4 CONVERGENCE

It is known that all importance sampling methods converge to the stationary point of the original objective (or to the optimum for a convex objective) as long as it is unbiased and the sampling probability is lower bounded. This is due to the fact that the convergence proof of most optimization methods like Sec 2.3 of (Défossez et al.) only requires the gradient estimator to be bounded and unbiased.

In our algorithm, we ensure the sampling probability is lower bounded by adopting probability smoothing. Furthermore, the sampling of our proposed method is unbiased, which many previous works on data selection have overlooked. These ensure the convergence of our proposed method.

## 4 RELATED WORK

In this section, we discuss previous work in static and dynamic data pruning.

**Static Data Pruning.**

Static data pruning chooses a fixed subset (coreset) of the whole training set throughout training. Static data pruning methods can be categorized as error based (Toneva et al., 2019; Paul et al., 2021), geometry based (Welling, 2009; Sener & Savarese, 2018), gradient matching based (Killamsetty et al., 2021a), and bilevel optimization methods (Killamsetty et al., 2021b;c).

Forgetting (Toneva et al., 2019) counts the number of forgetting events for each sample during training, samples that are more unforgettable are the pruned. GRAND and EL2N (Paul et al., 2021) statically prune samples based on the gradient norm and the L2-norm of the error. Herding (Welling, 2009) incrementally and greedily add samples that minimize the distance between the coreset center and the center of the original training set. K-Center greedy (Sener & Savarese, 2018) is a greedy approximation to the minimax problem, where the coreset is selected to minimize the largest distance of each sample to the closest coreset sample. GradMatch (Killamsetty et al., 2021a) select the coreset that approximates the gradient of the full training set. Lastly, Glister (Killamsetty et al., 2021b) and Retrieve (Killamsetty et al., 2021c) treat static data pruning as a bilevel optimization problem.

**Dynamic Data Pruning.**

Dynamic data pruning is also known as data importance sampling. Since the variance of an unbiased gradient estimator can be connected with the expected reduction of the loss function, the optimal sampling distribution to minimize the variance for sampling with replacement has been studied. Alain et al. (2015) and Zhao & Zhang (2015) derive that in the case of sampling with replacement, the optimal sampling probability of each sample is proportional to its gradient norm $\|\nabla \ell_i(\boldsymbol{\theta}_t)\|$. Alain et al. (2015) propose a distributed setting where $\|\nabla \ell_i(\boldsymbol{\theta}_t)\|$ is computed by the worker nodes. Zhao & Zhang (2015) propose to use the smoothness constant of the per-sample loss to approximate

$\|\nabla \ell_i(\boldsymbol{\theta}_t)\|$. Katharopoulos & Fleuret (2018) propose to use the gradient norm of the logits as an approximation, which requires a forward pass on unselected samples.

Despite the theoretical justification, sampling with replacement could result in repeated samples, which can be undesirable. Instead, researchers have started to propose other dynamic data pruning methods that avoid repeated samples. Jiang et al. (2019) propose to use the forward pass to obtain the loss of each sample and conduct Bernoulli sampling, where the sampling probability of each sample is determined by the ranking of its loss. Qin et al. (2023) propose to use the previous loss function value as the score to eliminate the additional forward pass. Samples with scores larger than the average are selected with probability 1, while the rest are selected with a discounted probability $r$. Raju et al. (2021) propose to maintain a moving average of the previous loss function value as the score. A fixed fraction of $F$ lowest score samples are selected with probability $\epsilon$, while the rest are selected with probability 1. Thao Nguyen et al. (2024) propose to prune a fraction of $F_i$ of the lowest-loss samples, where $F_i$ is determined by the epoch number $i$.

## 5 EXPERIMENTAL RESULTS

We compare the performance of our method with previous algorithms on both image classification and LLM fine-tuning tasks. For image classification, we consider CIFAR-10, CIFAR-100, and ImageNet-100. For LLM, we evaluate the performance of instruction finetuning LLaMA-7B using the Open LLM Leaderboard.

The CIFAR10/100 experiments are conducted on one NVIDIA RTX 2080Ti GPU with 11GB memory. The ImageNet-100 and LLaMA-7B experiments are conducted on one NVIDIA RTX A6000 GPU with 48GB memory.

Since Qin et al. (2023) already show that static pruning methods in general perform worse than dynamic pruning methods, we only compare our method with other dynamic pruning methods. Specifically, we include the following algorithms in our comparisons:

- Weighted Replacement (Katharopoulos & Fleuret, 2018): Sampling with replacement. The sampling weights are proportional to the gradient norm of the logits, which is obtained from an additional forward pass.

- Selective Backprop (Jiang et al., 2019): Bernoulli sampling. The sampling weights are decided by the ranking of the loss obtained from an additional forward pass.

- Extra Forward Bernoulli: Bernoulli sampling. The sampling weights follow our derived sampling distribution. Uses the gradient norm of the logits, which is obtained from an additional forward pass.

- Infobatch (Qin et al., 2023): Samples whose previous loss is lower than the average are selected by a discounted probability.

- Historical Replacement: Our modified version of Weighted Replacement (Katharopoulos & Fleuret, 2018) that utilizes historical statistics.

- Historical Ranking: Our modified version of Selective Backprop (Jiang et al., 2019) that utilizes historical statistics.

- Reducing Epoch: Reduces the number of epochs without any pruning.

- Reducing Epoch (scaled): Reduces the number of epochs without any pruning and increases the learning rate by the ratio of the reduced epochs.

- Random: Uniform sampling without replacement.

For Omission, Historical Replacement, and Historical Ranking, we select $\tilde{\beta}_1 = 0.9$, $\tilde{\beta}_2 = 0.99$, follow the commonly used parameters in Adam. For the score smoothing factor, we set $\alpha = 0.1$ for all the experiments, similar to what is commonly used for label smoothing..

Additionally, as discussed in Section 3.3, Infobatch (Qin et al., 2023) and our proposed algorithm (Omission) both sample from all of the $n$ training samples at the beginning of the epoch. The rest of the methods sample from a larger batch of $B = 2b$ samples. This is also the sampling ratio we use for Random. We observe that using a lower sampling ratio as in some previous work might result in a weaker baseline.

Table 1: Comparison of dynamic data pruning methods with extra forward on CIFAR-10/100.

| Dataset | CIFAR10 | | | CIFAR100 | | |
|---|---|---|---|---|---|---|
| Selection Ratio | 10 | 20 | 50 | 10 | 20 | 50 |
| Weighted Replacement (Katharopoulos & Fleuret, 2018) | $92.83_{\pm0.10}$ | $94.50_{\pm0.11}$ | $95.20_{\pm0.06}$ | $71.76_{\pm0.14}$ | $75.05_{\pm0.18}$ | $77.96_{\pm0.27}$ |
| Selective Backprop (Jiang et al., 2019) | $93.57_{\pm0.09}$ | $94.62_{\pm0.06}$ | $95.32_{\pm0.15}$ | $\mathbf{73.70}_{\pm0.26}$ | $76.09_{\pm0.35}$ | $77.78_{\pm0.22}$ |
| Extra Forward Bernoulli | $\mathbf{93.83}_{\pm0.11}$ | $\mathbf{94.77}_{\pm0.12}$ | $\mathbf{95.49}_{\pm0.14}$ | $73.68_{\pm0.08}$ | $\mathbf{76.18}_{\pm0.09}$ | $\mathbf{78.05}_{\pm0.10}$ |

These methods are compared in two groups. The first group utilizes an extra forward to obtain statistics at the current step, while the second group only utilizes historical statistics. In our experiments, we observe that the overhead for the second group is negligible. In addition, we observe that Reducing Epoch is an often neglected baseline that performs quite well. While Reducing Epoch (scaled) is compared in Infobatch (Qin et al., 2023), in our experiments, we observe that Reducing Epoch in general performs better than Reducing Epoch (scaled).

## 5.1 IMAGE CLASSIFICATION

We conduct image classification experiments on CIFAR-10/100 with ResNet18. The whole data is trained for 200 epochs using SGD with a batch size of 128. We use `drop_last=True` to drop the last batch of an epoch, as the size of the last batch is significantly smaller, which is hurting the performance for the baseline methods. We use the cosine annealing learning rate schedule with a maximum learning rate of 0.1, weight decay of 0.0005, and 10% warmup. Following the standard training procedure, we apply RandomHorizontalFlip, RandomCrop, and normalization for data augmentation. The results are averaged over four trials.

From Table 1 and Table 2, we can see that our proposed method has the best performance for both the extra forward and the historical setting. Additionally, Figure 1 and Figure 2 show the curve for training loss and validation accuracy over the wall clock time. From these curves, we can also observe that Omission has the best performance.

Table 2: Comparison of dynamic data pruning methods with historical statistics on CIFAR-10/100.

| Dataset | CIFAR10 | | | CIFAR100 | | |
|---|---|---|---|---|---|---|
| Selection Ratio | 10 | 20 | 50 | 10 | 20 | 50 |
| Reducing Epoch | $92.91_{\pm0.12}$ | $94.18_{\pm0.11}$ | $95.07_{\pm0.18}$ | $73.41_{\pm0.19}$ | $75.65_{\pm0.08}$ | $77.61_{\pm0.09}$ |
| Reducing Epoch (scaled) | $89.79_{\pm0.13}$ | $92.87_{\pm0.13}$ | $94.98_{\pm0.15}$ | $67.63_{\pm0.34}$ | $74.73_{\pm0.18}$ | $77.51_{\pm0.33}$ |
| Random | $92.82_{\pm0.06}$ | $94.21_{\pm0.03}$ | $95.12_{\pm0.12}$ | $73.14_{\pm0.32}$ | $75.49_{\pm0.14}$ | $77.51_{\pm0.11}$ |
| Infobatch (Qin et al., 2023) | $92.97_{\pm0.09}$ | $94.21_{\pm0.15}$ | $95.11_{\pm0.16}$ | $72.99_{\pm0.28}$ | $75.34_{\pm0.10}$ | $77.37_{\pm0.25}$ |
| Historical Replacement | $92.10_{\pm0.13}$ | $93.94_{\pm0.06}$ | $95.03_{\pm0.14}$ | $71.04_{\pm0.15}$ | $74.74_{\pm0.16}$ | $77.39_{\pm0.16}$ |
| Historical Ranking | $\mathbf{93.38}_{\pm0.08}$ | $94.35_{\pm0.10}$ | $95.27_{\pm0.13}$ | $72.82_{\pm0.21}$ | $75.32_{\pm0.07}$ | $77.03_{\pm0.21}$ |
| Omission | $93.23_{\pm0.11}$ | $\mathbf{94.60}_{\pm0.14}$ | $\mathbf{95.38}_{\pm0.12}$ | $\mathbf{73.47}_{\pm0.19}$ | $\mathbf{75.81}_{\pm0.13}$ | $\mathbf{77.99}_{\pm0.18}$ |
| Full Training | $95.47_{\pm0.07}$ | | | $78.77_{\pm0.22}$ | | |

Our experiment on ImageNet-100 is conducted with ResNet50. The data augmentation includes RandomHorizontalFlip, RandomResizedCrop, and normalization. The other settings follow the CIFAR-10/100 experiment.

From Table 3, we can see that our proposed method acehives the best performance on ImageNet-100. When using only 10%/20% of the samples, we are able to outperform Infobatch by 0.63%/0.42% accuracy respectively.

## 5.2 INSTRUCTION FINETUNING

We conduct experiments on instruction finetuning LLaMA-7B and use the Open LLM Leaderboard (Beeching et al., 2023) to evaluate the performance. We follow Infobatch (Qin et al., 2023) to select 1k high-quality samples from the Alpaca data set (Taori et al., 2023). We adopt the version where the response is generated by GPT-4. We train for 250 steps using AdamW with a batch size of 64. A

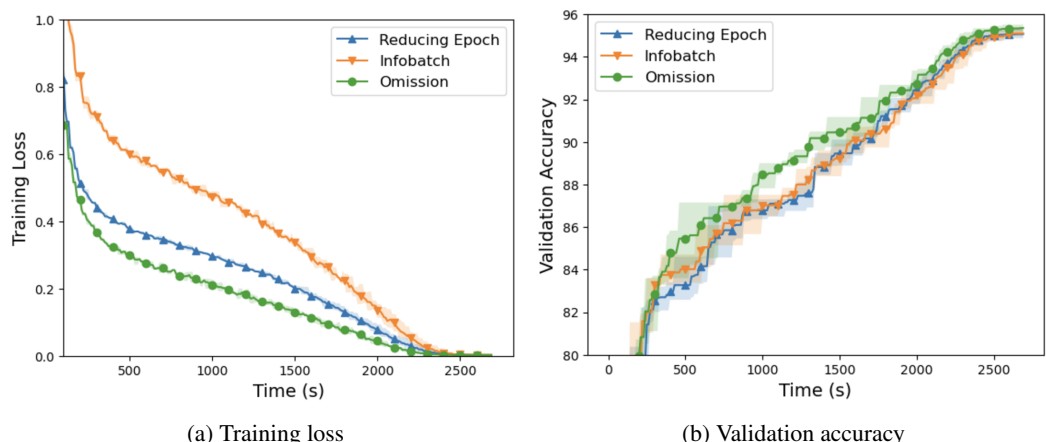

(a) Training loss

(b) Validation accuracy

Figure 1: Training loss and validation accuracy on CIFAR-10 for $50\%$ pruning ratio. The solid line shows the mean of the trials, while the shadow shows the entire value range across the trials. Infobatch has higher training loss as its gradient estimation is biased by a scalar.

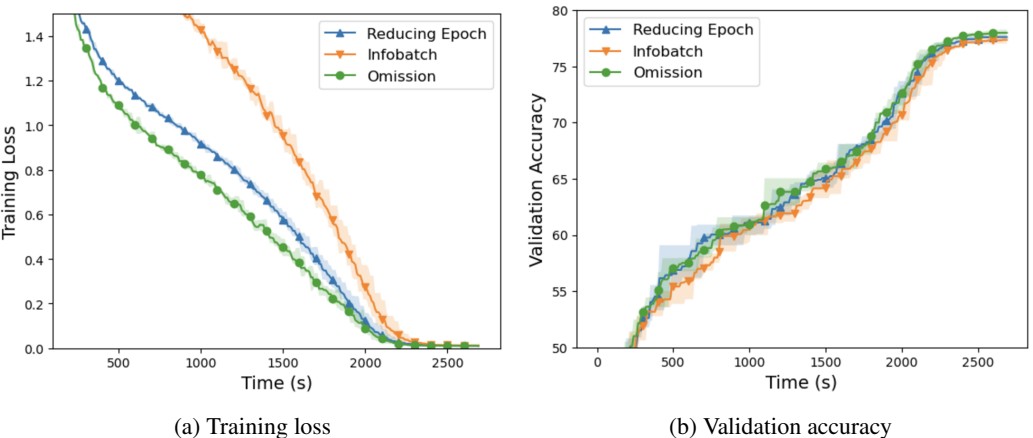

(a) Training loss

(b) Validation accuracy

Figure 2: Training loss and validation accuracy on CIFAR-100 for $50\%$ pruning ratio. The solid line shows the mean of the trials, while the shadow shows the entire value range across the trials. Infobatch has higher training loss as its gradient estimation is biased by a scalar.

linear decay learning rate schedule with a maximum learning rate of $0.00015$ is used. The results are averaged over four trials.

Table 4 shows that our proposed method improves the accuracy on the Open LLM Leaderboard for LLaMA-7B instruction finetuning. Since in this case the training data is not directly related to the test tasks, we observe that although our method acheives better training loss, it may not lead to better test performance on some tasks. However, our method still performs the best in 4 over 6 tasks, and achieves a better average performance.

## 5.3 ABLATION STUDY

We conduct ablation study on Cifar10 and Cifar100 to validate the importance of each component of the proposed algorithm. The results are shown in Table 5. As we can see, probability smoothing is the most important. Without probability smoothing, it can lead to very small sampling weights on Cifar10, causing worse performance and much larger variance. The tracking of the exponential moving average is the second most important, but without it, we still beat Infobatch, the previous state-of-the-art that does not have the exponential moving average. The least important is scale normalization, which still contributes to some performance improvement.

Table 3: Comparison of dynamic data pruning methods on ImageNet-100.

| Dataset | ImageNet-100 | | |
|---|---|---|---|
| Selection Ratio | 10 | 20 | 50 |
| Reducing Epoch | $81.72_{\pm 0.33}$ | $85.09_{\pm 0.13}$ | $86.84_{\pm 0.13}$ |
| Infobatch (Qin et al., 2023) | $80.96_{\pm 0.11}$ | $84.77_{\pm 0.31}$ | $86.89_{\pm 0.24}$ |
| Omission | $\mathbf{82.35}_{\pm 0.27}$ | $\mathbf{85.51}_{\pm 0.12}$ | $\mathbf{87.18}_{\pm 0.11}$ |
| Full Training | $87.41_{\pm 0.08}$ | | |

Table 4: Comparison of dynamic data pruning methods on the Open LLM Leaderboard.

| Dataset | ARC | HellaSwag | GSM8K | MMLU | TruthfulQA | Winogrande | Avg. |
|---|---|---|---|---|---|---|---|
| No Data Selection | $53.75_{\pm 0.27}$ | $\mathbf{79.72}_{\pm 0.04}$ | $10.79_{\pm 0.49}$ | $36.29_{\pm 0.22}$ | $37.71_{\pm 0.40}$ | $71.35_{\pm 0.14}$ | $48.27_{\pm 0.06}$ |
| Infobatch (Qin et al., 2023) | $53.65_{\pm 0.43}$ | $79.69_{\pm 0.06}$ | $10.90_{\pm 0.37}$ | $\mathbf{36.45}_{\pm 0.08}$ | $37.57_{\pm 0.35}$ | $71.31_{\pm 0.25}$ | $48.26_{\pm 0.04}$ |
| Omission | $\mathbf{54.12}_{\pm 0.21}$ | $79.64_{\pm 0.05}$ | $\mathbf{11.03}_{\pm 0.39}$ | $36.15_{\pm 0.18}$ | $\mathbf{37.94}_{\pm 0.27}$ | $\mathbf{71.43}_{\pm 0.21}$ | $\mathbf{48.39}_{\pm 0.11}$ |

Table 5: Ablation on CIFAR-10/100.

| Dataset | CIFAR10 | | | CIFAR100 | | |
|---|---|---|---|---|---|---|
| Selection Ratio | 10 | 20 | 50 | 10 | 20 | 50 |
| Omission | $93.23_{\pm;0.11}$ | $\mathbf{94.60}_{\pm;0.14}$ | $\mathbf{95.38}_{\pm;0.12}$ | $\mathbf{73.47}_{\pm;0.19}$ | $\mathbf{75.81}_{\pm;0.13}$ | $\mathbf{77.99}_{\pm;0.18}$ |
| Omission (no ema) | $93.07_{\pm;0.01}$ | $94.41_{\pm;0.06}$ | $95.16_{\pm;0.10}$ | $73.22_{\pm;0.17}$ | $75.45_{\pm;0.36}$ | $77.48_{\pm;0.40}$ |
| Omission (no normalization) | $\mathbf{93.26}_{\pm;0.13}$ | $94.54_{\pm;0.07}$ | $95.33_{\pm;0.13}$ | $73.06_{\pm;0.07}$ | $75.48_{\pm;0.15}$ | $\mathbf{77.99}_{\pm;0.13}$ |
| Omission (no smoothing) | $90.95_{\pm;1.51}$ | $88.45_{\pm;5.41}$ | $92.84_{\pm;1.95}$ | $73.22_{\pm;0.36}$ | $75.49_{\pm;0.26}$ | $77.82_{\pm;0.41}$ |
| Infobatch | $92.97_{\pm;0.09}$ | $94.21_{\pm;0.15}$ | $95.11_{\pm;0.16}$ | $72.99_{\pm;0.28}$ | $75.34_{\pm;0.10}$ | $77.37_{\pm;0.25}$ |

## 6 CONCLUSIONS

Noticing that most dynamic data pruning methods that avoid repeated samples can be viewed as weighted Bernoulli sampling, in this work, we derive the optimal distribution to minimize variance, thereby accelerating convergence. We then propose Omission (**O**ptimal inspired historical we**i**ghted Bern**o**ulli sampli**n**g), which tracks the exponential moving average of the squared gradient norm and adopts scale normalization as well as probability smoothing to improve the performance. Experiments on image classification and large language model instruction finetuning tasks show that our proposed method achieves higher accuracy than other dynamic pruning methods.

**Limitations and Future Work.** Even though we derive the optimal sampling distribution to minimize the variance of weighted Bernoulli sampling given a fixed sampling budget, the optimal sampling budget for each epoch is still unknown. While we adopt a fixed $50\%$ sampling ratio in our experiments, methods with dynamic sampling ratios have been proposed. For example, the annealing mechanism proposed by Infobatch (Qin et al., 2023) disables dynamic data pruning near the end of the training. This can be seen as a way to set different sampling budgets for each epoch, which can be combined with our proposed method. We leave the topic of setting a dynamic sampling ratio for future exploration.

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

# A  APPENDIX

## A.1  PROOF OF THEOREM 1.

Given the independence of each Bernoulli sample, the variance is simply the sum of the variances of the individual samples. For the $i$-th sample, the variance is given by

$$p_i(1 - p_i)\frac{\|\boldsymbol{g}_i\|^2}{B^2 p_i^2} = (1 - p_i)\frac{\|\boldsymbol{g}_i\|^2}{B^2 p_i}.$$

Taking the derivative of this expression, we obtain $-\|\boldsymbol{g}_i\|^2/(B^2 p_i^2)$, whose absolute value is monotonically decreasing.

For any sampling probability distribution $p_1, \ldots, p_B$ satisfying the budget constraint $\sum_{i=1}^{B} p_i = b$, if there exist a sample pair $(i, j)$ such that

$$\frac{\|\boldsymbol{g}_i\|^2}{B^2 p_i^2} > \frac{\|\boldsymbol{g}_j\|^2}{B^2 p_j^2}$$

and $p_i < 1$, then we can add a suitably small $\delta$ to $p_i$ and subtract $\delta$ from $p_j$ to reduce the variance.

This is because the derivative of the variance is $-\|\boldsymbol{g}_i\|^2/(B^2 p_i^2)$ with respect to $p_i$ and $-\|\boldsymbol{g}_j\|^2/(B^2 p_j^2)$ with respect to $p_j$. Thus, increasing $p_i$ by $\delta$ and decreasing $p_j$ by $\delta$ leads to change in variance of

$$[\frac{\|\boldsymbol{g}_j\|^2}{(B^2 p_j^2)} - \frac{\|\boldsymbol{g}_i\|^2}{(B^2 p_i^2)}]\delta,$$

which is negative according to our condition.

Consequently, for the optimal probability distribution $p_1^*, \ldots, p_B^*$, it should satisfy

$$\begin{cases} \frac{\|\boldsymbol{g}_i\|^2}{B^2 (p_i^*)^2} \geq d, & \text{if } p_i^* = 1 \\ \frac{\|\boldsymbol{g}_i\|^2}{B^2 (p_i^*)^2} = d, & \text{otherwise} \end{cases}$$

for all $i$ for some constant $d$.

Taking square roots on both sides and rearranging the equations completes the proof.

## A.2  PROOF OF THEOREM 2.

Similar to the proof of Theorem 1, we start by decomposing the variance into the sum of the variances of the individual samples.

Since

$$\mathbb{E}[\mathbb{1}\{r_i = 1\}\frac{\boldsymbol{g}_i}{p_i} - \boldsymbol{g}_i] = \boldsymbol{0},$$

we have

$$\mathbb{E}[\|\mathbb{1}\{r_i = 1\}\frac{\boldsymbol{g}_i}{p_i} - \mathbb{E}[\boldsymbol{g}_i]\|^2] = \mathbb{E}[\|\mathbb{1}\{r_i = 1\}\frac{\boldsymbol{g}_i}{p_i} - \boldsymbol{g}_i\|^2] + \mathbb{E}[\|\boldsymbol{g}_i - \mathbb{E}[\boldsymbol{g}_i]\|^2]$$

Consequently, for the $i$-th sample, the variance is given by

$$\mathbb{E}[p_i(1 - p_i)\frac{\|\boldsymbol{g}_i\|^2}{B^2 p_i^2}] + \mathbb{E}[\|\boldsymbol{g}_i - \mathbb{E}[\boldsymbol{g}_i]\|^2]$$

where the first term accounts for the variance of Bernoulli sampling, while the second term accounts for the variance of the random variation.

Taking the derivative of this expression, we obtain $-\mathbb{E}[\|\boldsymbol{g}_i\|^2]/(B^2 p_i^2)$, the rest just follows the proof of Theorem 1.

