# OpenReview forum: "Learn from the Past: Dynamic Data Pruning with Historically Weighted Bernoulli Sampling"
_ICLR.cc/2025/Conference — Submitted to ICLR 2025_

### Official Review · Reviewer_JqRJ · 2024-10-30

**Soundness:** 2
**Presentation:** 3
**Contribution:** 2
**Rating:** 5
**Confidence:** 4

**Summary:**

The paper focuses on improving dynamic data pruning or data importance sampling,
with the goal of enhancing the training efficiency of large machine learning
models by skipping less critical samples during training. The authors adopt
Bernoulli sampling to avoid repeated samples and derive the optimal distribution
to minimize variance. They propose a method called Omission for implementation.
Numerical experiments were conducted on image classification tasks (CIFAR-10,
CIFAR-100, ImageNet-100) and large language model (LLM) fine-tuning tasks.

**Strengths:**

Numerical experiments were conducted on image classification tasks and large language model fine-tuning.

**Weaknesses:**

The paper lacks novelty and originality. The optimal sampling distribution has
already been derived more rigorously and investigated more deeply in previous
works for various settings. Some relevant earlier references are listed below,
and there have been additional developments since then:

- Ting, D., Brochu, E. Optimal subsampling with influence functions. *Advances in Neural Information Processing Systems*, 2018; 31.
- Wang, H., Zou, J. A comparative study on sampling with replacement vs Poisson sampling in optimal subsampling. In *International Conference on Artificial Intelligence and Statistics*, 2021 Mar 18 (pp. 289-297). PMLR.

The statement "in practice we can keep sampling until a fixed batch size is
reached" is scientifically incorrect. Doing so changes the distribution of the
selected samples, rendering all theoretical results invalid.

Notations are poorly defined, leading to many confusing statements. For example,
what are the relationships among $n$, $B$, and $b$? How does the batch variance
relate to the empirical loss? Additionally, from the second sentence of Section
3.2, it appears that you are only focusing on the logit loss, which could easily
confuse readers who are not already experts in the field. Clearer explanations
and definitions are needed to make the content more accessible.

**Questions:**

1. Do you mean an argument in an existing software package when you say "drop_last=True"?
2. What is the meaning of the numbers in the tables?
3. Why discuss Table 1 and Table 5 first, before the other tables? Why not move Table 5 to be Table 2?
4. What is the "Prune Ratio"?
5. The lines in the figures are not distinguishable in grayscale.

---

> ### Author Response · Authors · 2024-11-25
>
> We thank the reviewer for the feedback.
>
> > The paper lacks novelty and originality. The optimal sampling distribution has already been derived more rigorously and investigated more deeply in previous works for various settings. Some relevant earlier references are listed below, and there have been additional developments since then.
>
> We thank the reviewer for pointing us to these sources.
>
> The main contribution of this paper is to model the problem of data selection as finding the optimal sampling distribution and making it practical through adaptations, rather than simply deriving the optimal distribution itself.
>
> For the two papers you mentioned, they focus on the general problem of estimation through sampling. Their experiments focus on a one time static estimation where the statistics are available. In contrast, dynamic data selection has the following two prominent features:
>
> 1. Multi-turn selection: Dynamic data selection resamples data for every epoch instead of a simple one time selection.
> 2. Unavailability (Cost) of statistics: The cost of directly computing the gradients defeat the purpose of data selection.
>
> The second feature is one of the main challenges for dynamic data selection. In [1], they use an approximate gradient from the current epoch with a mechanism to estimate if the cost is worth it. For [2][3], they rely on stale information from previous samplings. In [2], they use the ranking of loss to decide the sampling probability, while [3] shows that a simple thresholding mechanism can also work.
>
> We argue that our novelty over these state-of-the-art methods lies in two parts. First, the awareness for the use of the optimal sampling distribution rather than some heuristics. Second, there is effort from us to make the method practical. As we can see from Table 5, without EMA, using stale information for the optimal sampling distribution actually does not have much improvement over Infobatch [2], a simple thresholding mechanism, while removing probability smoothing will make the performance a lot worse.
>
> [1] Katharopoulos, A., & Fleuret, F. (2018, July). Not all samples are created equal: Deep learning with importance sampling. In International conference on machine learning (pp. 2525-2534). PMLR.
>
> [2] Jiang, A. H., Wong, D. L. K., Zhou, G., Andersen, D. G., Dean, J., Ganger, G. R., ... & Pillai, P. (2019). Accelerating deep learning by focusing on the biggest losers. arXiv preprint arXiv:1910.00762.
>
> [3] Qin, Z., Wang, K., Zheng, Z., Gu, J., Peng, X., Zhou, D., ... & You, Y. InfoBatch: Lossless Training Speed Up by Unbiased Dynamic Data Pruning. In The Twelfth International Conference on Learning Representations.
>
> > The statement "in practice we can keep sampling until a fixed batch size is reached" is scientifically incorrect. Doing so changes the distribution of the selected samples, rendering all theoretical results invalid.
>
> We are sorry for the ambiguity of the sentence and we have revised it in the paper. As mentioned in Sec 3.3, what we really mean by that statement is the following: Yes, they are theoretically different. However, in practice we observe their performance to be similar as shown in the following table, where we compare the version of Omission with/without fixed batch size:
>
> | Dataset                          |            | Cifar10       |           |          |Cifar100           |           |
> | -------------------------------- | ----------------- | ----------------- |----------------- |----------------- |----------------- |----------------- |
> | Selection Ratio                  | 10                | 20                | 50                | 10                | 20                | 50                |
> | Omission (fixed batch size)      | 93.23$\\pm$0.11 | 94.60$\\pm$0.14 | 95.38$\\pm$0.12 | 73.47$\\pm$0.19 | 75.81$\\pm$0.13 | 77.99$\\pm$0.18 |
> | Omission (varying batch size) | 93.28$\\pm$0.10 | 94.51$\\pm$0.13 | 95.35$\\pm$0.07 | 73.46$\\pm$0.06 | 75.96$\\pm$0.22 | 77.97$\\pm$0.27 |
>
> As mentioned in the paper, a huge part of this work is to make the optimal distribution “inspired” sampling practical. In the process, there are many approximations being made without affecting the convergence such as using the approximated gradient and the EMA on stale information. This is yet another approximation to make our method more useful. Specifically, this makes the method easier to use (the sampling logic can be more separated in code), while also being slightly faster due to the use of a fixed batch size.
>
> We have revised Sec 3.3 to make it more clear in the paper.

---

> > ### Author Response · Authors · 2024-11-25
> >
> > > Notations are poorly defined, leading to many confusing statements. For example, what are the relationships among n, B, and b?
> >
> > These are defined in the earlier section of the paper, but we agree we can recall definitions when they are used in later sections. At the start of Sec 2, we define n to be the total number of samples in the training set. In Sec 3, we mention that B is the size of the original batch that you want to approximate, while b is the number of samples you choose to approximate that batch.
> >
> > > How does the batch variance relate to the empirical loss?
> >
> > We mention the relation between batch gradient variance and empirical loss in equation (2). As one can see from equation (2), the variance of the batched gradient contributes to the expected reduction of the empirical loss. Thus, by reducing the variance, one can expect a faster reduction in the empirical loss.
> >
> >
> > > Additionally, from the second sentence of Section 3.2, it appears that you are only focusing on the logit loss, which could easily confuse readers who are not already experts in the field. Clearer explanations and definitions are needed to make the content more accessible.
> >
> > Yes, we will improve this part in our revision. For clarity, we like to point out that this approach is general (logits are just the last layer output of the neural network) and is not limited by the choice of the loss function. Additionally, this is a common approximation made in data selection papers to make them more practical.
> >
> > > Do you mean an argument in an existing software package when you say "drop_last=True"?
> >
> > Yes, it is used for dropping the last batch of an epoch. This is because the last epoch of an epoch during Cifar training is significantly smaller, which is hurting the performance of the baseline methods. We have made this clear in the revision.
> >
> > > What is the meaning of the numbers in the tables?
> >
> > It is the validation accuracy. The percentage of correct predictions on the validation set.
> >
> > > Why discuss Table 1 and Table 5 first, before the other tables? Why not move Table 5 to be Table 2?
> >
> > This is a typo, so we have fixed the typo without changing the ordering of the tables. Table 1 and Table 2 are the main results, while Table 5 is for the ablation study.
> >
> > > What is the "Prune Ratio"?
> >
> > This is a term also used by previous data selection papers for the portion of samples that are not selected during an epoch.
> >
> > However, this is a typo in our paper as we are actually referring to the selection ratio, which is the portion of samples that are selected during an epoch. We have fixed this in the revision.
> >
> > > The lines in the figures are not distinguishable in grayscale.
> >
> > Thanks for the suggestion. We have improved this in the revision.
> >
> > We thank the reviewer again for the detailed feedback.

---

> > > ### Author Response · Authors · 2024-12-02
> > >
> > > Dear Reviewer JqRJ,
> > >
> > > We believe we have answered your question about novelty and all the other questions.
> > > Could you consider adjusting your score if our response addresses your concerns?
> > >
> > > Our novelty lies in
> > > 1. Modeling the problem of data selection as finding the optimal sampling distribution
> > > 2. Proposing a practical sampling method that is inspired by the optimal sampling distribution.
> > >
> > > Most existing work on data selection misses this important connection to optimal sampling distribution and proposes simple heuristics.
> > > Furthermore, directly using the optimal sampling distribution is not practical due to the unique challenges of data selection. We propose several mechanisms in order to make the proposed optimal-sampling-inspired method practical. Otherwise, using stale information for the optimal sampling distribution actually does not have much improvement over simple heuristics.
> > >
> > > We are happy to answer any additional questions from the reviewers.
> > >
> > > All authors of Submission 8998

---

### Official Review · Reviewer_NJ1s · 2024-10-31

**Soundness:** 3
**Presentation:** 3
**Contribution:** 3
**Rating:** 6
**Confidence:** 4

**Summary:**

This paper studies dynamic data pruning, also known as data importance sampling, which has been proposed to improve the efficiency of training large machine learning models. In Section 2, Background, the reason for dynamic sampling is well explained by establishing the inequality (2). The main point of this paper is to propose a new way to choose p_i as in Theorem 1 (similarly in Theorem 2) so that a smaller upper bound in inequality (2) can be achieved. The rest of the paper details how to implement the formula in Theorem 1 and 2.

**Strengths:**

Overall, a well written paper. It's easy to follow the main idea.

**Weaknesses:**

Section 3.2.1 EXPONENTIAL MOVING AVERAGE OF GRADIENT NORM SQUARED is not clearly written. This reviewer has difficulty to see the value of this subsubsection. It should be better motivated.

In theorem 1 and 2, the determination of p_i requires ||g_i||, for all i=1,\ldots, n (or B), which may not be realistic. It seems that this will be the limitation of the proposed approach. The authors tend to address it in the paper. However, there doesn't seem to be a convincing solution. If one can't compute ||g_i|| for all observations (samples), then the proposed method doesn't seem sensible. Nevertheless, I am unsure how we should consider this in our evaluation...

**Questions:**

No

---

> ### Author Response · Authors · 2024-11-25
>
> We thank the reviewer for the positive feedback!
>
> >  Section 3.2.1 EXPONENTIAL MOVING AVERAGE OF GRADIENT NORM SQUARED is not clearly written. This reviewer has difficulty seeing the value of this subsubsection. It should be better motivated.
>
> The motivation of this section is that to avoid the cost for data selection one has to use historical data. However, historical data can be noisy and largely varying in scale. Thus, unlike [1][2], which simply uses the data from the last sampling, we propose to use the exponential moving average on the squared gradient norm with scale normalization.
>
> We have revised Sec 3.2.1 to make it clearer.
>
> [1] Jiang, A. H., Wong, D. L. K., Zhou, G., Andersen, D. G., Dean, J., Ganger, G. R., ... & Pillai, P. (2019). Accelerating deep learning by focusing on the biggest losers. arXiv preprint arXiv:1910.00762.
>
> [2] Qin, Z., Wang, K., Zheng, Z., Gu, J., Peng, X., Zhou, D., ... & You, Y. InfoBatch: Lossless Training Speed Up by Unbiased Dynamic Data Pruning. In The Twelfth International Conference on Learning Representations.
>
> > In theorem 1 and 2, the determination of p_i requires ||g_i||, for all i=1,\ldots, n (or B), which may not be realistic. It seems that this will be the limitation of the proposed approach. The authors tend to address it in the paper. However, there doesn't seem to be a convincing solution. If one can't compute ||g_i|| for all observations (samples), then the proposed method doesn't seem sensible. Nevertheless, I am unsure how we should consider this in our evaluation…
>
> Yes, this is one of the challenges we aim to solve in the paper. To make it practical, we use the historical data with an exponential moving average to avoid the computational cost. Additionally, the adoption of probability smoothing not only ensures biasedness, but also ensures that everyone is being sampled with some frequency so that the historical data will not be too out-dated.

---

> > ### Author Response · Authors · 2024-12-02
> >
> > Dear Reviewer NJ1s,
> >
> > Thanks again for your constructive feedback and writing suggestions!
> >
> > We have improved our writing based on your suggestions. We believe this will greatly help the reader to understand the motivation and the underlying mechanism of our proposed method.
> >
> > Would you like to increase your score to reflect the improvement made upon your suggestions?
> > We are also happy to answer any additional questions from the reviewers.
> >
> > All authors of Submission 8998

---

### Official Review · Reviewer_TybE · 2024-11-03

**Soundness:** 3
**Presentation:** 4
**Contribution:** 2
**Rating:** 6
**Confidence:** 2

**Summary:**

This paper proposes a weighted Bernoulli sampling method, called Omission, for dynamic data pruning, which can minimize the variance of the gradient estimate without the problem of repeatedly selected samples in sampling with replacement. Aside from this theoretical motivation, the method Omission is also made more practically robuste and implementable through techniques of score smoothing, score normalisation, and estimation of the sample gradient squared norm from historical statistics. The provided experimental results on tasks of image classification and LLM instruction finetuning attest to the competitive performance of Omission. An ablation study was carried out to investigate the importance of the practical techniques used in the implementation of Omission.

**Strengths:**

* The proposition of a dynamic data pruning method that yields an unbiased gradient estimate of minimal variance while avoiding the selection of repeated samples is a valuable theoretical contribution.

* In addition to the solide theoretical foundation, several practical techniques are employed for a more efficient and robust implementation of the proposed weighted Bernoulli sampling scheme.

* An extensive empirical study is provided, where the proposed algorithm is observed to perform competitively against a range of related algorithms.

**Weaknesses:**

* Considering that the empirical results are reported over only four trials, the observed performance gains of Omission, which are not always significantly large with respected to the error bars, may be induced by the randomness of trials.

**Questions:**

* Would the empirical performance gains of Omission remain stable when the number of trials is increased?

* Why is the performance of full training not provided for the experiments reported in Table 4? And what is the pruning ratio used in these  experiments?

---

> ### Author Response · Authors · 2024-11-25
>
> Thanks for the positive feedback!
>
> > The observed performance gains of Omission, which are not always significantly large with respect to the error bars, may be induced by the randomness of trials.
> > Would the empirical performance gains of Omission remain stable when the number of trials is increased?
>
> Yes, the performance gains of Omission remain stable when the number of trials is increased.
>
> We show the result of 12 trials in the following table.
>
> | Dataset         |    | Cifar10  |       |   |   Cifar100   |      |
> | --------------- | --------------- | --------------- |--------------- |--------------- |--------------- |--------------- |
> | Selection Ratio | 10              | 20              | 50 | 10 | 20 | 50 |
> | Reducing Epoch  | 92.88$\\pm$0.11 | 94.23$\\pm$0.12 | 95.09$\\pm$0.15 | 73.42$\\pm$0.18 | 75.55$\\pm$0.12 | 77.62$\\pm$0.07 |
> | Infobatch       | 92.95$\\pm$0.09 | 94.23$\\pm$0.13 | 95.10$\\pm$0.13 | 73.00$\\pm$0.24 | 75.35$\\pm$0.11 | 77.32$\\pm$0.26 |
> | Omission        | 93.18$\\pm$0.13 | 94.61$\\pm$0.13 | 95.38$\\pm$0.11 | 73.48$\\pm$0.18 | 75.80$\\pm$0.14 | 77.95$\\pm$0.20 |
>
> > Why is the performance of full training not provided for the experiments reported in Table 4? And what is the pruning ratio used in these experiments?
>
> Thank you for the suggestion.
>
> It is worth noticing that full training is the same as the baseline (“Reducing Epoch”, which does not employ any data selection) but simply with a different number of training steps.
>
> For Cifar10 and Cifar100, people usually define full training as 100 epochs. However, for the experiment in Table 4, there is no such an establishment.
>
> Instead, we select the training steps that are the best for the baseline (250 steps) and also use it for Infobatch and Omission for comparison. As we can see from the following table, increasing the number of training steps actually degrades the baseline performance.
>
> We rename the “Reducing Epoch” in Table 4 to “No Data Selection” to improve clarity. Thanks again for the suggestion.
>
>
> |                                  | ARC             | hellaswag       | gsm8k           | mmlu            | truthfulqa (0-shot) | winogrande            | Avg             |
> | -------------------------------- | --------------- | --------------- | --------------- | --------------- | ------------------- | --------------------- | --------------- |
> | No Data Selection (250 steps) | 53.75$\\pm$0.27 | 79.72$\\pm$0.04 | 10.79$\\pm$0.49 | 36.29$\\pm$0.22 | 37.71$\\pm$0.40     | 71.35$\\pm$0.14| 48.27$\\pm$0.06 |
> | No Data Selection (500 steps)    | 51.56$\\pm$0.45 | 79.52$\\pm$0.01 | 8.34$\\pm$0.76  | 36.38$\\pm$0.44 | 37.11$\\pm$0.52     | 70.07$\\pm$0.19       | 47.15$\\pm$0.06 |
> | No Data Selection (1000 steps)   | 47.86$\\pm$1.09 | 77.36$\\pm$0.05 | 5.38$\\pm$0.16  | 34.88$\\pm$0.78 | 39.12$\\pm$1.57     | 65.77$\\pm$0.60       | 45.06$\\pm$0.46 |
>
> As for the sampling ratio, we follow Infobatch [1] and set it to 50%, which chooses 50% of the samples for each pass of the training set.
>
> [1] Qin, Z., Wang, K., Zheng, Z., Gu, J., Peng, X., Zhou, D., ... & You, Y. InfoBatch: Lossless Training Speed Up by Unbiased Dynamic Data Pruning. In The Twelfth International Conference on Learning Representations.

---

> > ### Comment · Reviewer_TybE · 2024-12-02
> >
> > I thank the authors for their reply. However after reading the other reviews (specially the one by Reviewer JqRJ) and the authors' responses, I am having some reservation about this paper. As someone not familiar with the literature on data sampling, I was unaware of the two references pointed out by Reviewer JqRJ, which seems important and very related to the present work. A thorough literature review should have already included these references in my opinion.
> >
> > As my initial positive rating was largely based on the interest of the results in Theorem 1 and Theorem 2, I will decrease my score to 6. I will also drop my confidence level to reflect my unfamiliarity with the field.

---

> > > ### Author Response · Authors · 2024-12-03
> > >
> > > Dear Reviewer TybE,
> > >
> > > We thank the reviewer for the response. We understand your potential concern about the two references pointed out by Reviewer JqRJ.
> > >
> > > We believe Reviewer JqRJ excels in a research direction (sampling & estimation) that is different but closely related to dynamic data selection. We are delighted that Reviewer JqRJ brought these two papers to our notice.
> > >
> > > We view this as an opportunity to bring these results to the awareness of the broader data selection community. As far as we know, before this there seems to be little interaction between the two directions. Even the latest papers like [1][2] utilize simple heuristics.
> > >
> > > Our contributions lie in realizing the connection between dynamic data selection and the optimal sampling distribution. Additionally, we propose several mechanisms in order to make the proposed optimal-sampling-inspired method practical. Otherwise, using stale information for the optimal sampling distribution actually does not have much improvement over simple heuristics.
> > >
> > > [1] Jiang, A. H., Wong, D. L. K., Zhou, G., Andersen, D. G., Dean, J., Ganger, G. R., ... & Pillai, P. (2019). Accelerating deep learning by focusing on the biggest losers. arXiv preprint arXiv:1910.00762.
> > >
> > > [2] Qin, Z., Wang, K., Zheng, Z., Gu, J., Peng, X., Zhou, D., ... & You, Y. InfoBatch: Lossless Training Speed Up by Unbiased Dynamic Data Pruning. In The Twelfth International Conference on Learning Representations (ICLR 2024).
> > >
> > > Thanks again for the response.
> > > We are happy to answer any additional questions from the reviewers.
> > >
> > > All authors of Submission 8998

---

### Comment · Area_Chair_bYZ6 · 2024-11-23
**Authors engagement**

Dear Authors,

The discussion period will close in a few days, but we have yet to hear from you. I recommend that you address the reviewers comments, as it will help the reviewers and me evaluate your work better and provide a well-informed recommendation to the SAC and PCs.

Best,

Your AC.

---

> ### Author Response · Authors · 2024-11-23
>
> Dear AC,
>
> We are in the middle of writing the rebuttal response. Experiments on LLM is slightly time consuming due to our limited resources. Thanks for your patience!
>
> Sincerely yours,
> Submission8998 Authors

---

### Meta-Review · Area_Chair_bYZ6 · 2024-12-19

**Metareview:**

The paper proposes a new algorithm for sampling training examples in the context of training large machine-learning models. Reviewer JqRJ highlighted that more general results exist and that Theorems 1 and 2 do not present novel material. Additionally, the empirical results demonstrate only minimal improvements over the baseline methods used for comparison.

For a paper of this nature, the bar for acceptance is higher and typically requires either novel theoretical contributions or strong, replicable performance improvements in practical applications. This paper does not meet that standard in its current form.

**Additional Comments On Reviewer Discussion:**

The authors and reviewers engaged and two of them were not convinced by the authors. Also once of them sent me a long feedback why the paper should not be accepted. I agree with his recommendation.

---

### Decision · Program_Chairs · 2025-01-22

Reject